# A model of resource partitioning between foraging bees based on learning

**Thibault Dubois**[1,2☯]*, **Cristian Pasquaretta**[1☯], **Andrew B. Barron**[2], **Jacques Gautrais**[1], **Mathieu Lihoreau**[1]

**1** Research Centre on Animal Cognition (CRCA), Centre for Integrative Biology (CBI); CNRS, University Paul Sabatier–Toulouse III, Toulouse, France, **2** Department of Biological Sciences, Macquarie University, Sydney, Australia

☯ These authors contributed equally to this work.
* thibault.dubois@univ-tlse3.fr

## Abstract

Central place foraging pollinators tend to develop multi-destination routes (traplines) to exploit patchily distributed plant resources. While the formation of traplines by individual pollinators has been studied in detail, how populations of foragers use resources in a common area is an open question, difficult to address experimentally. We explored conditions for the emergence of resource partitioning among traplining bees using agent-based models built from experimental data of bumblebees foraging on artificial flowers. In the models, bees learn to develop routes as a consequence of feedback loops that change their probabilities of moving between flowers. While a positive reinforcement of movements leading to rewarding flowers is sufficient for the emergence of resource partitioning when flowers are evenly distributed, the addition of a negative reinforcement of movements leading to unrewarding flowers is necessary when flowers are patchily distributed. In environments with more complex spatial structures, the negative experiences of individual bees on flowers favour spatial segregation and efficient collective foraging. Our study fills a major gap in modelling pollinator behaviour and constitutes a unique tool to guide future experimental programs.

## Author summary

Pollinating animals, like bees, face the challenge of maximising their returns on plant resources while minimising their foraging costs. Observations show bees establish idiosyncratic foraging routes (traplines) to visit familiar plants using short paths. This is an effective strategy for collecting pollen and nectar that are dispersed and renewable resources. Intriguingly, different bees seem to establish non-overlapping traplines, which aids in partitioning resources. So far, however, how bees establish these foraging strategies is a mystery. It seems unfeasible for them to be able to negotiate with competing foragers. Here we present a simple computational model derived from empirical observations suggesting bees can develop efficient routes between flowers while minimizing spatial overlaps with competitors based only on their history of reinforcement in a floral array. In the model, bees learn to return to flowers where they found nectar and avoid flowers that

**Data Availability Statement:** The code which generated our data is fully available at the following URL: https://gitlab.com/jgautrais/resourcepartitioninginbees/-/releases. It allows for

generation of similar data as the one used in our paper.

**Funding:** TD was funded by a co-tutelle PhD grant from the University Paul Sabatier (Toulouse) and Macquarie University (Sydney). ABB was supported by the Templeton World Charity Foundation project grant TWCF0266. CP and ML were supported by research grants of the Agence Nationale de la Recherche (ANR-16-CE02-0002-01, ANR-19-CE37-0024, ANR-20-ERC8-0004-01) to ML. The funders had no role in study design, data collection and analysis, decision to publish, or preparation of the manuscript.

**Competing interests:** The authors have declared that no competing interests exist.

were found empty. Numerical simulations of our model predict the emergence of resource partitioning between pairs of bees under various conditions. This suggests a simple strategy to promote efficient foraging among competing agents on a renewable resource that could apply to many different pollinating animals.

## Introduction

Foraging animals are expected to self-distribute on food resources in order to minimize competition and maximize their individual net energy gain [1,2]. Resource partitioning between individuals of different species is well documented, and often results from functional [3,4] or behavioural [5,6] specializations. By contrast, how individuals of the same species interact to exploit resources in a common foraging area is less understood [7,8].

For pollinators, such as bees that individually exploit patchily distributed floral resources in environments with high competition pressure, efficient resource partitioning appears a prodigious problem to solve. It involves assessing the quality of food resources, their spatial distribution, their replenishment rate, and the activity of other pollinators. As central place foragers, bees often visit familiar feeding sites (plants or flower patches) in a stable sequence called a "trapline" [9,10]. Individual bees with exclusive access to an array of artificial flowers tend to develop traplines minimizing travel distances to visit all the necessary flowers to fill their nectar crop and return to the nest (e.g. bumblebees: [11–13]; honey bees: [14]). This routing behaviour involves spatial memories that can persist days [15] or weeks [16].

How bees partition resources, when several conspecifics exploit the same foraging area, is however an open question. Experimentally the problem is challenging to address as it requires monitoring the movements of numerous bees simultaneously over large spatial and temporal scales. In theory, bees should develop individualistic traplines that minimize travel distances and spatial overlap with other foragers, thereby improving their own foraging efficiency and minimizing the costs of competition [17,18]. Best available data supporting this hypothesis come from observations of small numbers of bumblebees foraging on potted plants [19,20] or artificial flowers (in effect mimicking inflorescences or plants) [18,21] in large flight tents. In these experimental foraging conditions with limited numbers of bees and feeding sites, foragers tend to avoid spatial overlaps as a consequence of competition by exploitation (when bees visited empty flowers) and interference (when bees interacted on flowers) [21].

Computational modelling is a powerful approach to further explore the mechanisms by which such partitioning might emerge from spatial learning and competitive interactions. At the individual level, trapline formation has been modelled using an iterative improvement algorithm where a bee compares the net length of the route it has just travelled (sum of the lengths of all transitions between two flowers, or the nest and a flower, comprising the flower visitation sequence) to the length of the shortest route experienced so far [22]. If the new route is shorter (or equivalent), the bee increases its probability of using all the transitions composing this route in its subsequent foraging bout. After several iterations, this route-based learning heuristic typically leads to the discovery and selection of a short (if not the shortest possible) trapline, thereby replicating observations in bees across a wide range of experimental conditions [23]. Note however that this model makes the strong assumption that bees can compute, memorize and compare the lengths of multiple routes upon return to their nest. To address this issue, it was proposed that trapline formation could also emerge from vector-based learning [24], in which the bee remembers independent vectors instead of complete routes. This form of learning was thought to be more parsimonious and plausible considering the current

understanding of spatial computation in the insect brain [25]. So far, however, none of these traplining algorithms have accounted for social interactions and current models that include bee foraging either did not consider individual specificities of movements based on learning and memory [26–30], or implemented them very succinctly without being the focus of the model [31]. Thus presently, there has been no formal exploration of how resource partitioning between interacting bees might form.

Here, we investigated the behavioural mechanisms underlying resource partitioning among traplining bees by comparing predictions of three agent-based models to each other. The different models integrate learning behaviour and social interactions in slightly different ways. Recent work showed that resource partitioning in bats foraging on patchily distributed cacti can be explained by basic reinforcement rules, so that a bat that finds an abundant feeding site tends to return to this site more often than its conspecifics [32]. Since bees extensively rely on associative learnings to recognize flowers and develop foraging preferences [33], we hypothesized that the combination of positive experiences (when a flower is full of nectar) and negative experiences (when a flower is unrewarding) could lead to the emergence of resource partitioning when different bees learn to use spatially segregated routes [18,21]. First, we developed models implementing biologically plausible navigation (derived from vector-based learning) based on positive and negative reinforcements of transition probabilities between flowers and tested the independent and combined influences of these feedback loops on trapline formation by comparing simulations to published experimental data. Next, we explored how these simple learning rules at the individual level can promote complex patterns of resource partitioning at the collective level, using simulations with multiple foragers in environments with different resource distributions.

## Results

### Overview of models

We designed models of agents (bees) foraging simultaneously in a common set of feeding sites (flowers) from a central location (colony nest) (see summary in Fig 1). A full description of the models is available in the ODD protocol (S1 Text). Briefly, each bee completes a succession of foraging trips (foraging bouts) defined as the set of movements and flower visits between the moment it leaves the nest until the moment it returns to it. Each bee initially moves between the different flowers using a distance-based probability matrix [22,23]. The probability to move between each flower is then modulated each time the bee finds the flower rewarding (positive reinforcement) or unrewarding (negative reinforcement). Learning occurs after each flower visit (online learning). We implemented three models to explore different combinations of positive and negative reinforcements: model 1: positive reinforcement only (hereafter "Model 1[+]"), model 2: negative reinforcement only (Model 2[–]), model 3: positive and negative reinforcements (Model 3[+/-]). Model comparison thus informed about the effect of each of the rules separately and in combination.

### Simulations with one forager

We first tested the ability of our models to replicate trapline formation by real bees, by comparing simulations of a single forager to published experimental data in which individual bumblebees were observed developing traplines between five equally rewarding artificial flowers in a large open field [13,22]. Lihoreau *et al.* [22] tested seven bumblebees in a regular pentagonal array (S1A Fig), which we judged enough to run quantitative comparisons with model simulations. While Woodgate *et al.* [13] tested six bees in a narrow pentagonal array (S1B Fig), only three of them presented enough successive foraging bouts in a single day to allow statistical

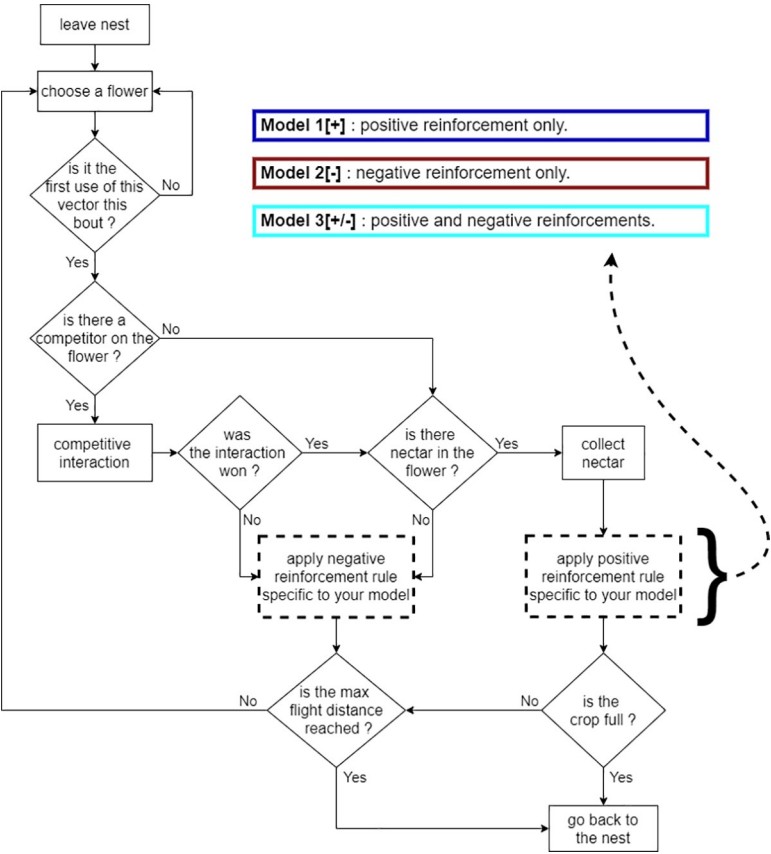

**Fig 1. Flowchart summarizing the agent-based models.** Rectangles represent actions performed by a bee. Diamonds indicate conditional statements. Arrows connect the different modules. The dashed rectangles are subject to the different rules of the three models.

comparisons with our model. Thus, for this dataset only qualitative comparisons were made with the model simulations. All statistical results are presented in Table 1.

We assessed the ability of bees to develop efficient routes by computing an index of route quality (i.e. the squared volume of nectar gathered divided by the distance travelled; see Methods). For real bees, route quality increased significantly with time in the regular pentagonal array of flowers (Fig 2A). When comparing simulations to experimental data, there were no

**Table 1. Statistical output for simulations with one individual.** Comparisons of route quality and route similarity through Binomial GLMMs using bee identity as a random effect (bee identity nested in simulation identity for simulated data).

| Variable | Data | Estimate | P |
|---|---|---|---|
| Route Quality | Exp. Data (Intercept) | 0.153 ± 0.023 | 0.001 |
| | Model 1[+] | -0.027 ± 0.023 | 0.224 |
| | Model 2[−] | -0.155 ± 0.023 | 0.001 |
| | Model 3[+/-] | -0.022 ± 0.023 | 0.339 |
| Route Similarity | Exp. Data (Intercept) | 0.110 ± 0.020 | 0.001 |
| | Model 1[+] | 0.088 ± 0.020 | 0.001 |
| | Model 2[−] | -0.109 ± 0.020 | 0.001 |
| | Model 3[+/-] | 0.086 ± 0.020 | 0.001 |

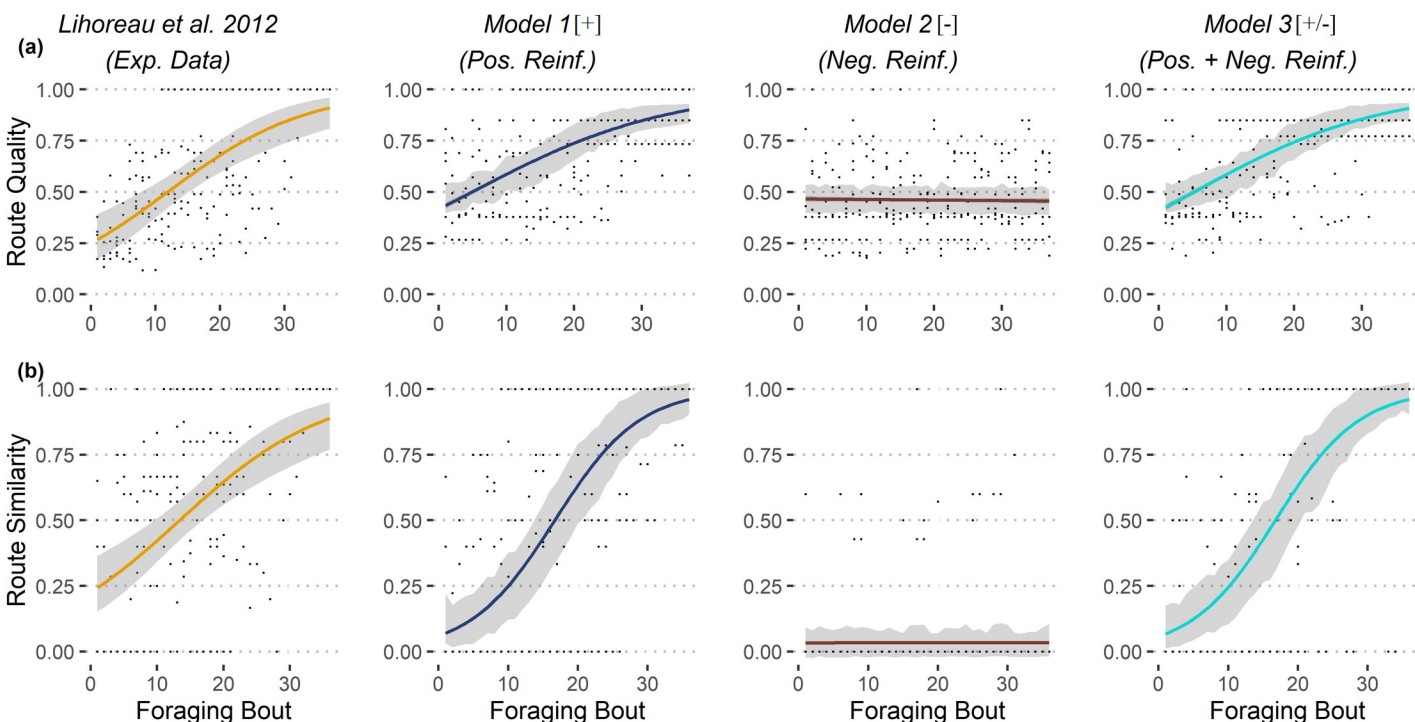

**Fig 2. Comparisons of experimental and simulated route qualities.** Comparisons of route qualities (a) and route similarities (b) between simulations and experimental data (regular pentagonal array of flowers as in [22]). See details of models in Fig 1. For each dataset, we show the estimated average trends across foraging bouts (coloured curves), along the standard error (grey areas) of the mean. For the sake of eye comparison, in the simulation plots the standard error of the mean is estimated from a sample of 7 simulations (N = 7 bees in [22]). Average trends were estimated over 500 simulation runs, using GLMM Binomial model with bee identity as random effect (bee identity nested in simulation identity for simulated data).

significant differences in trends with models 1[+] and 3[+/-] (Table 1), meaning that simulated bees developed routes of similar qualities as real bees. However, route qualities predicted by model 2[−] were significantly lower than the experimental data. Similar trends were observed in the narrow pentagonal array of flowers (S3 Text).

We assessed the ability of bees to develop stable routes using an index of route similarity (i.e. computing the number and percentage of transitions between two flowers (or the nest and a flower) shared between two successive routes; see Methods). Route similarity is set between 0 (the two routes are completely different) and 1 (the two routes are completely identical). For real bees, route similarity increased with time in the regular pentagonal array (Fig 2B). When comparing simulations to experimental data, route similarity increased significantly more in models 1[+] and 3[+/-] than for real bees. However, route similarity in model 2[−] was significantly lower than for real bees. Similar trends were observed in the narrow pentagonal array (S3 Text).

Thus overall, positive reinforcement is necessary and sufficient to replicate the behavioural observations (although with a significant difference found for route similarity between the experimental data and the models 1[+] and 3[+/-]), while negative reinforcement has no detectable effect.

## Simulations with two foragers

Having tested our models with one forager, we next explored conditions for the emergence of resource partitioning within pairs of foragers. Here experimental data are not available for

**Table 2. Statistical output for simulations with two individuals.** Comparisons of (i) exploitation competition, (ii) interference competition, (iii) route similarity, (iv) route partitioning and (v) collective foraging efficiency through GLMMs using bee identity as a random effect (bee identity nested in simulation identity for simulated data). The results presented are the slope estimate along with a 95% confidence interval of the mean, for each type of environment tested (See Methods for details).

| Variable | Data | Estimate (1 patch) | Estimate (2 patches) | Estimate (3 patches) |
|---|---|---|---|---|
| Exploitation Competition | Model 1[+] | -4.26e-03 ± 2.10e-04 | 6.27e-03 ± 1.80e-04 | 6.65e-03 ± 1.90e-04 |
| | Model 2[−] | -3.32e-03 ± 1.90e-04 | -2.10e-02 ± 2.00e-04 | -2.06e-02 ± 2.00e-04 |
| | Model 3[+/-] | -8.94e-03 ± 2.20e-04 | -1.88e-02 ± 3.00e-04 | -1.05e-02 ± 2.00e-04 |
| Interference Competition | Model 1[+] | -4.57e-03 ± 7.20e-04 | 1.05e-02 ± 4.00e-04 | 9.16e-03 ± 5.20e-04 |
| | Model 2[−] | -2.49e-03 ± 7.40e-04 | -2.10e-02 ± 6.00e-04 | -1.68e-02 ± 7.00e-04 |
| | Model 3[+/-] | -1.53e-02 ± 8.00e-04 | -1.66e-02 ± 7.00e-04 | -1.01e-02 ± 6.00e-04 |
| Route Similarity | Model 1[+] | 1.34e-01 ± 2.00-e03 | 9.56e-02 ± 1.30e-03 | 7.76e-02 ± 1.20e-03 |
| | Model 2[−] | 7.46e-04 ± 6.65e-03 | 1.91e-02 ± 2.90e-03 | -3.20e-02 ± 3.10e-03 |
| | Model 3[+/-] | 1.33e-01 ± 2.00e-03 | 6.95e-02 ± 1.20e-03 | 6.14e-02 ± 1.30e-03 |
| Route Partitioning | Model 1[+] | 2.90e-02 ± 1.30e-03 | -1.02e-02 ± 1.30e-03 | -8.26e-03 ± 1.26e-03 |
| | Model 2[−] | 1.22e-02 ± 1.30e-03 | 1.28e-02 ± 1.20e-03 | 1.82e-02 ± 1.20e-03 |
| | Model 3[+/-] | 3.55e-02 ± 1.30e-03 | 3.17e-02 ± 1.30e-03 | 2.19e-02 ± 1.30e-03 |
| Collective Foraging Efficiency | Model 1[+] | 4.20e-02 ± 1.50e-03 | -4.61e-03 ± 1.27e-03 | 3.04e-03 ± 1.25e-03 |
| | Model 2[−] | -5.08e-03 ± 1.24e-03 | -8.03e-03 ± 1.27e-03 | -4.24e-03 ± 1.24e-03 |
| | Model 3[+/-] | 4.12e-02 ± 1.50e-03 | 8.77e-03 ± 1.25e-03 | 1.83e-02 ± 1.30e-03 |

comparison. We thus simulated foraging patterns and interactions of bees in different types of environments defined by flower patchiness. Each environment contained 10 flowers that were either distributed in one patch, two patches, or three patches (see examples in S2 Fig; for details, see Methods). Each bee had to visit five rewarding flowers to fill its crop to capacity. All the statistical results of this part are presented in Table 2.

**Exploitation and interference competition.** We first analysed exploitation competition by quantifying the frequency of visits to non-rewarding flowers by each bee during each foraging bout. The frequency of visits to non-rewarding flowers decreased for simulated bees in models 2[−] and 3[+/-] (Fig 3A and Table 2), irrespective of the environment they were tested in. However, in model 1[+], bees behaved differently in the different environments. In the one patch environment, bees decreased their visits to non-rewarding flowers, whereas in the two and three patch environments, bees tended to increase their visits to non-rewarding flowers. The increase of non-rewarding visits in environments with patchily distributed resources can be explained as follows. If bees start reinforcing visits to flowers of a shared patch, they will become more likely to visit the same patch. Given the much larger space between flowers of different patches than between flowers of the same patch, the probability to switch from one patch to the next (without the help of the negative reinforcement) is low, leading to bees flying between the empty flowers of a patch repeatedly. This process ultimately increases visits to empty flowers, and also occurrences of interference between the two bees if they are both at the same patch.

We analysed interference competition by quantifying the number of interactions on flowers at each foraging bout between the two bees. The frequency of encounters on flowers decreased with time for both models 2[−] and 3[+/-] (Fig 3B and Table 2), irrespective of the type of environment. Here again, bees of model 1[+] behaved differently in the different environments. In the one patch environment, bees decreased their frequency of encounters on flowers, whereas in the two and three patches environments they increased their frequency of interactions. Again this is likely due to the absence of negative reinforcement, leading bees to be trapped in an empty patch.

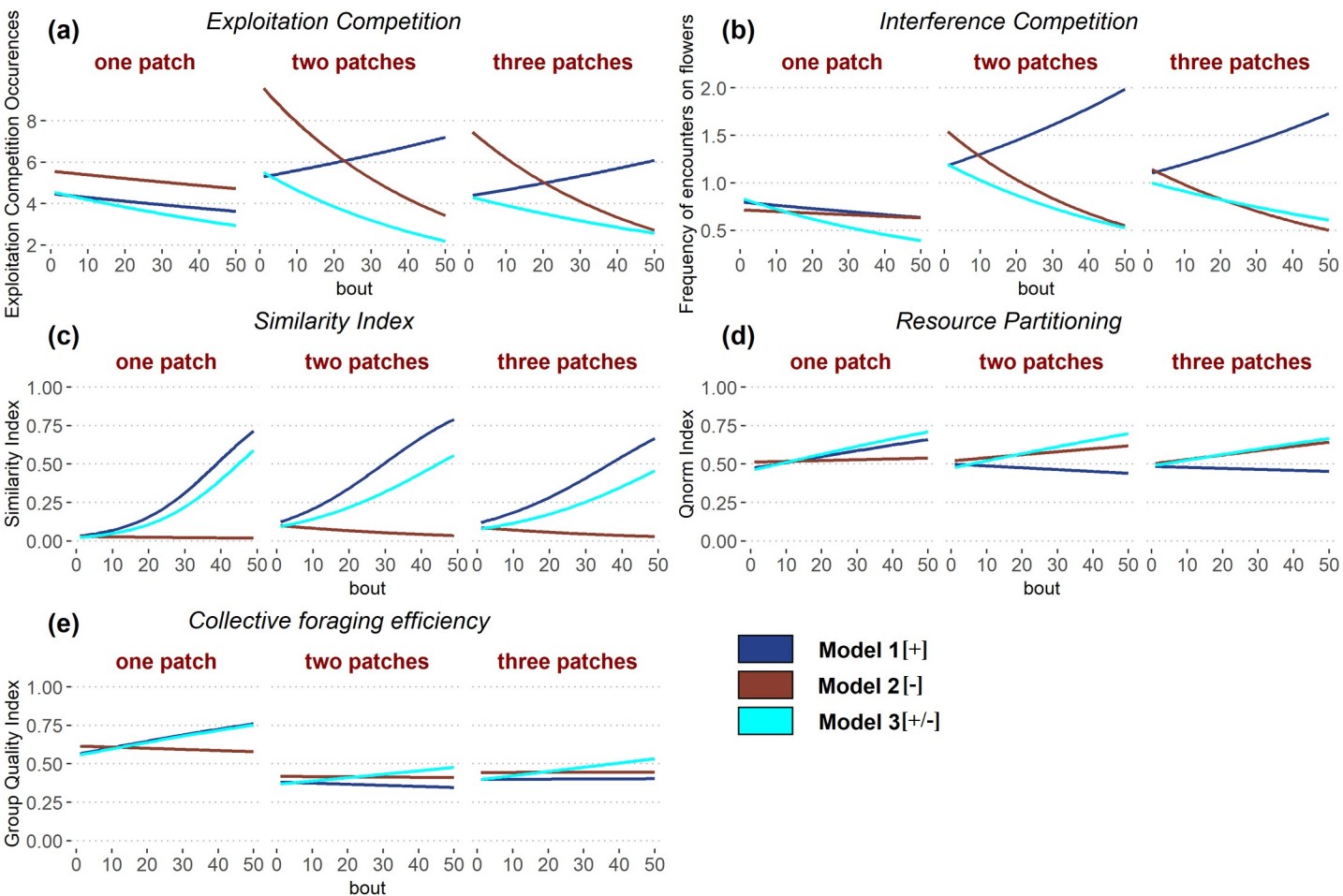

**Fig 3. Model comparisons for observed variables.** Results of simulations with two foragers in environments with 10 flowers. See details of models in Fig 1. The x axis is the number of completed foraging bouts by the two foragers. The y axis represents respectively: (a) the estimated mean frequency of visits to empty flowers; (b) the estimated mean frequency of encounters on flowers; (c) the similarity index $SI_{ab}$ between two successive flower visitation sequences; (d) the index of resource partitioning $Q_{norm}$ (0: both bees visit the same set of flowers; 1: bees do not visit any flower in common); (e) the collective foraging efficiency index $QL_{group}$. Average trends for each model are estimated across all types of environments (one patch, two patches and three patches; see S2 Fig).

These differences in the occurrence of exploitation and interference competition correlate to a variation in the total number of visits to flowers, effectively improving the bees' foraging efficiency. The strength of this effect is greater for the exploitation competition as it is occurring much more often (exploitation: 2 to 10 occurrences in average; interference: 0 to 2 occurrences in average; Fig 3A and 3B).

Thus, overall negative reinforcement was necessary for reducing exploitation and interference competition. By allowing bees to avoid empty flowers, negative reinforcement facilitated the discovery of new flowers and thus gradually relaxed competition. In the absence of negative reinforcement, both types of competition increased in environments with several flower patches.

**Route similarity.** We analysed the tendency of bees to develop repeated routes by comparing the similarity between flower visitation sequences of consecutive foraging bouts for each individual (Fig 3C). Bees increased route similarity through time in all types of environments in models 1[+] and 3[+/-] (Table 2). By contrast, in model 2[−], route similarity did not vary in the one patch environment and decreased through time in the other environments.

The presence of negative reinforcement in models 2[–] and 3[+/-] reduced the final level of route similarity compared to trends found in model 1[+]. In these conditions, bees learned to avoid revisits to empty flowers and showed greater variation in their visitation sequences as a result of searching for new flowers.

**Resource partitioning.**   We analysed the level of resource partitioning by quantifying the tendency of the two bees to use different flowers. This index varies between 0 (the two bees use the same set of flowers) and 1 (the two bees use completely different sets of flowers; see Methods).

In model 1[+], bees showed an increase of resource partitioning with time in environments with one patch, and a decrease in environments with two or three patches (Fig 3D and Table 2). By contrast, in model 2[–] and model 3[+/-], bees showed an increase of resource partitioning with time in all types of environments. Model 3[+/-] displayed similar levels of partitioning in all the different environments where models 1[+] and 2[–] showed a greater variance. Model 1[+] had greater partitioning only in the one patch environment, while model 2[–] had greater partitioning in the two and three patch environments. This suggests positive and negative reinforcements contributed unevenly but complementarily in the model 3[+/-] with different spatial distributions of flowers. Positive reinforcement would be the main driver for partitioning in the one patch environment, while negative reinforcement would be the main driver in the two and three patches environments.

**Collective foraging efficiency.**   To quantify the collective foraging efficiency of bees, we analysed the capacity of the two foragers to reach the most efficient combination of route qualities (i.e. minimum distance travelled by a pair of bees needed to visit the 10 flowers; see Methods).

In model 1[+], pairs of bees increased their collective foraging efficiency with time in environments with one and three patches (Fig 3E and Table 2). By contrast, bees decreased their level of foraging efficiency in the environment with two patches. In model 2[–] pairs of bees decreased their collective foraging efficiency with time in all types of environments. In model 3[+/-] bees increased their collective foraging efficiency with time in all types of environments. Positive reinforcement seems to be the main driver for collective foraging efficiency in the one patch environment. However, neither the positive or negative reinforcements alone managed to increase foraging efficiency in the two and three patch environments. Only their interaction, as seen in the model 3[+/-], brought an increase in collective foraging efficiency. Collective efficiency is generally higher in the one patch environment than in the two and three patches environments because the difference between the best possible path (for which the collective foraging efficiency is equal to 1) and a typical suboptimal path of a simulated bee is lower due to the absence of long inter-patch movements.

## Discussion

Central place foraging animals exploiting patchily distributed resources that replenish over time are expected to develop foraging routes (traplines) minimizing travel distances and interactions with competitors [17,18,34]. Here we developed cognitively plausible agent-based models of probabilistic navigation to explore the behavioural mechanisms leading to resource partitioning between traplining bees. In the models, bees learn to develop routes as a consequence of feedback loops that modify the probabilities of moving between flowers. Simulations show that, in environments where resources are evenly distributed, bees can reach high levels of resource partitioning based on positive reinforcement only, but cannot do so based on negative reinforcement only. However, in environments with patchily distributed resources, both positive and negative reinforcements become necessary.

We developed our hypotheses and models based on observations on single foraging bees [13,22]. Our first step was therefore to test how the models compared to existing data. Models with positive reinforcement showed a good general fit to the experimental data (Fig 2 and Fig A in S3 Text), although they often overestimated the increase of route similarity with experience in real bees. This imperfect match could be due to the low amount of available experimental data in the original studies (seven individuals in [22], three individuals in [13]), but also a result from the limitations of our models. First, the model bees navigate with the only intent of finding resources, while real bees sometimes show phases of stochastic exploration during and after the trapline formation [13,35]. Second, real bees do not always find a flower when exploring their environment, especially when naïve. On the contrary, there is no probability of not finding a flower for the model's bees, which then visit on average more flowers from the first foraging bout. The resulting routes are of higher route quality as they visit more different flowers, but of lower similarity as they also use different transitions between flowers, while real bees navigate back and forth between the same few flowers.

We then used our models for predicting behaviours of two competing bees in different types of environments. To develop a trapline in this competitive situation, the bees needed to find rewarding flowers but also avoid competitors. These two goals were independently fulfilled by the positive reinforcement and the negative reinforcement. Simulated bees were fastest to develop a trapline when using the positive reinforcement only, and unable to follow any stable route when solely using the negative reinforcement. However, this does not indicate that the use of both reinforcements was less effective than just positive reinforcement. Simulated bees were indeed more likely to establish a stable route with positive reinforcement only, but these routes most likely contained contested flowers that the bees were not able to give up on, as they did not change their behaviour after experiencing unrewarded visits. This assumption is supported by the fact that both reinforcements (model 3[+\-]) leads to a greater resource partitioning and a higher collective foraging efficiency.

When foraging in uniformly distributed plant resources (one patch), it is easiest to encounter all the resources available as none of them are isolated far from any other (with thus a low probability of being reached). Consequently, two bees are very likely over time to learn non-overlapping foraging routes and show resource partitioning. However, in environments with non-uniformly distributed resources (two or three patches), the added spatial complexity can interfere with this process. The initial likelihood of moving between distant patches is relatively low. Thus, the sole implementation of positive reinforcement often does not allow bees to explore all possible patches, so that the paths of competing bees overlap and interfere within a subset of the available patches. Adding a negative reinforcement for movement transitions leading to unrewarded flowers increases aversion for these empty flowers, the spatial segregation of foraging paths between competing bees and the collective exploitation of all available patches, even if the initial probabilities of moving to distant patches are low. This interplay between the influences of positive and negative experiences at flowers on the spatial and competitive decisions of bees is in accordance with the behavioural observations that bees tend to return to rewarding flowers and avoid unrewarding flowers, either because flowers were found empty or because the bees were displaced by a competitor during a physical encounter [18,21].

The need for a negative reinforcement to enhance discrimination between different options or stimuli is well-known in learning theory and behavioural studies [36–38]. At the individual level, negative experiences modulate learning. For both honey bees and bumblebees, adding negative reinforcement to a learning paradigm (e.g. quinine or salt in sucrose solution) enhances fine scale colour discrimination [39] and performance in cognitive tasks requiring learning of rules of non-elemental associations [40]. The insect brain contains multiple distinct neuromodulatory systems that independently modulate negative and positive reinforcement

[41] and the ability of bees to learn negative consequences is well-established [42]. At the collective level, negative feedbacks are also known to modulate social and competitive interactions. This is especially notable in collective decisions making by groups of animals and robots [43], where negative feedbacks enable individuals to make fast and flexible decisions in response to changing environments [44,45]. Even so, the utility of negative reinforcement to enhance efficient trapline formation and the consequences of this for the emergence of effective resource partitioning has not been commented on previously. It may be that this is a general phenomenon with applicability to other resource partitioning systems.

Our study implies that some very basic learning and interaction rules are sufficient for trapline formation and resource partitioning to emerge in bee populations, providing a solid basis for future experimental work. Nonetheless, several improvements of the model could already be considered for further theoretical investigations of bee spatial foraging movements and interactions. These could include adding to the model the documented inter-individual variability in cognitive abilities [46,47] and spatial strategies [48] of bees, the variability in the nutritional quality of resources [49,50] and the specific needs of each colony [51], or the well-known ability of bees to use chemical [52], visual [53] and social information to decide whether to visit or avoid flowers. For instance, foragers of many bee species leave chemical cues as footprints on the flowers they have visited (bumblebees and honeybees: [54]; solitary bees: [55]). Bees learn to associate the presence or absence of a reward to these footprints and to revisit or avoid scented flowers [56]. Such a pheromone system is a beneficial signal for all participants in the interaction [54]. This additional information could significantly enhance the positive or negative experiences of bees visiting flowers and thus increase resource partitioning to the benefit of all bees coexisting in the patch (S4 Text). Even different species of bee can learn to use these cues [54,57]. More exploration could also be done in the future in regards to the probability of winning a competitive interaction on flower. While we considered all individuals to have similar probabilities to access floral nectar when bees encounter on flowers, resource partitioning has been suggested to be favoured by asymmetries in foraging experiences [11,18]. Differences in experience or motivation would ultimately affect the outcome of competition, both passively (more consistent depletion of the flowers in a trapline) and actively (active displacement of other bees from one's established trapline).

Our study fills a major gap in our understanding of pollinator behaviour and interactions by building on recent attempts to simulate trapline foraging by individual bees [22–24]. It constitutes a unique theoretical modelling framework to explore the spatial foraging movements and interactions of bees in ecologically relevant conditions within and between plant patches, thereby holding considerable premises to guide novel experiments. Further developments of the model could be used to test predictions with more than two bees (see examples S1 Video and S4 Text), several colonies, or even different species of bees (e.g. honey bees) and thus complement current predictions about pollinator population dynamics [29–31]. Ultimately, the robust predictions of the spatial movements and interactions of bees over large spatio-temporal scales, through experimental validations of the model, have the potential to show the influence of bee movements on plant reproduction patterns and pollination efficiency [58,59].

## Methods

### Description of models

We built three agent-based models in which bees learn to develop routes in an array of flowers (see summary diagram in Fig 1). The environment contains flowers each delivering the same quality and volume of nectar. At each foraging bout (flower visitation sequence, beginning and ending at the colony nest entrance as the bee starts and finishes a foraging trip, respectively),

each bee attempts to collect nectar from five different flowers in order to fill its nectar crop (stomach) to capacity. Flowers are refilled between foraging bouts. In simulations with two bees, the two individuals start their foraging bout synchronously, and the flowers are refilled with nectar after the last bee has returned to the nest. For each bee, flower choice is described using movement transitions (orientated jumps between two flowers or between the nest and a flower). The initial probability of using each possible transition is based on the length of the movement, so that short transitions have a higher probability than longer ones. This probability is then modulated through learning when the bee used a transition for the first time during a bout.

We implemented two learning rules: (i) a positive reinforcement, i.e. if the flower at the end of a transition contains nectar and the bee feeds on it, it is set as a rewarding experience and the probability to reuse the transition later is increased; (ii) a negative reinforcement, i.e. if the flower is empty or if the bee is pushed away by competitors, it is set as a non-rewarding experience and the probability to reuse the transition later is decreased. The three models implemented either one of these two rules (model 1[+]: positive reinforcement only; model 2[–]: negative reinforcement only) or both rules (model 3[+/-]).

A flower is empty if it had previously been visited in the same foraging bout by the same or another bee (exploitation competition). If multiple bees visit a flower at the same time (interference competition), only one bee (randomly selected) is allowed to stay and take the reward if there is one. The other bees react as if the flower was empty. After each flower visit, all bees update their probabilities to reuse the movement transitions accordingly.

Trapline formation thus depends on the experience of the bee and its interactions with other foragers. For simplicity, we restricted our analysis to two bees. Working with pairs of bees facilitates future experimental tests of the models' predictions by reducing the number of bees to manipulate and control in experiments [11,18]. Note, however, that the same models can be used to simulate interactions among more bees (see examples with five bees in S1 Video and S4 Text).

A detailed description of the model is provided in S1 Text, in the form of an Overview, Design concepts and Details (ODD) protocol [60,61]. The complete R code is available at *https://gitlab.com/jgautrais/resourcepartitioninginbees/-/releases*.

## Environments

**Simulations with one forager.** Our first goal was to test the ability of our models to replicate observations of real bees. To do so, we ran simulations in environments replicating published experimental studies in which individual bumblebees (*Bombus terrestris*) were observed developing traplines between five equally rewarding artificial flowers in a large open field [13,22]. To our knowledge, these studies provide the most detailed available datasets on trapline formation by bees. Lihoreau *et al.* [22] used a regular pentagonal array of flowers (S1A Fig) in which they tracked seven bumblebees. We judged this sample size enough to run quantitative comparisons with model simulations (raw data are available in the supporting information of [22]). Woodgate *et al.* [13] used a narrow pentagonal array of flowers (S1B Fig). Here, however, the small sample size of the original dataset (three bumblebees, data shared by J. Woodgate) only enabled a qualitative comparison with the model simulations (S3 Text).

**Simulations with two foragers.** We then explored conditions leading to resource partitioning by running model simulations with two foragers. Here we simulated environments containing 10 flowers, in which each bee had to visit five rewarding flowers to fill its crop to capacity. The simulated flowers should thus be considered as feeding sites such as plants or inflorescences, which are more likely to contain such large amounts of resources (20% of the

bee's crop). To test whether model predictions were robust to variations in spatial distributions of resources we simulated three types of environments characterized by different levels of resource patchiness: (*i*) a patch of 10 flowers, (*ii*) two patches of five flowers each, and (*iii*) three patches of five, three and two flowers respectively (see examples in S2 Fig). We generated flower patches in a spatial configuration comparable to the one used in both experimental set-ups [13,22]. In a 500m x 500m plane, a nest was set as the centre (coordinates 0,0). Then, patch centres were placed with a minimum distance of 160m between each, and at least 20m from the nest. Within a patch, flowers were randomly distributed according to two constraints: (i) flowers were at least 20m apart from each other and from the nest, (ii) the maximum distance of each flower from the centre of their patch was 40m. This ensured that each patch had a maximum diameter of 80m and inter-flower distances were smaller between all flowers of the same patch than between all flowers of different patches (See ODD Protocol for more details, S1 Text, Ch.7 "Submodels"). The distances used in the simulated environments were chosen to replicate the experimental data used to test the model [13,22] where closest flowers were spaced by 25 metres. In our model, however, only the relative distance between the different elements of the environment mattered as all distances were normalized in the process of creating the probability matrix (S1 Text).

## Movements

At each step, a bee chooses to visit a target location (flower or nest) based on a matrix of movement probabilities. This matrix is initially defined using the inverse of the square distance between the current position of the bee and all possible target locations [22,23]. The probability of moving from location *i* to location *j* among multiple possible targets, is initially set to:

$$P(i \rightarrow j) = \frac{\frac{1}{d^2_{ij}}}{\sum_j \frac{1}{d^2_{ij}}} \tag{1}$$

Where $d_{ij}$ is the distance between locations *i* and *j*. The use of a movement probability matrix is justified by its capacity to approximate accurately the probability to reach a flower using a random walk, although it is significantly dependent on what exponent is used in the formula transforming distances to probabilities (See S6 Text for details). Thus, while the probability matrix allows unexperienced bees (during their first foraging bout) to move between all flowers, it should not be interpreted as a knowledge of their positions, but rather a probability of finding them by chance.

Before choosing its destination, the bee lists all possible target locations. For simplicity, the bee excludes its current location, thus preventing looping flights to and from the same target (flower or nest), which are rare in experienced bumblebee foragers [62] and provide little information about bee routing behaviour. The bee also excludes the location it had just come from to simulate the tendency of bumblebees to avoid recently visited (and thus depleted) flowers [62]. The foraging bout ends if: (*i*) the bee fills its crop to capacity, (*ii*) the bee chooses the nest as a target destination, or (*iii*) the bee reaches a maximum travelled distance of 3000 m. The latest was added to avoid endless foraging trips in the model. The maximum distance was chosen based on the observation that bumblebees typically forage within a distance of less than 1km from their nest [63–65].

## Learning

Learning modulates the probability of using transition movements as soon as the bee experiences the chosen target and only once within a foraging bout (the first time the transition is

used during the foraging bout; Fig 1). This approach has the advantage of implementing vector navigation [24,25] (S6 Text) and thus avoids assumptions about computation and comparison of complete routes [22,23], but it makes new assumptions about bees remembering a large number of locations and distances of flowers. Bees are known to be able to learn few independent feeding sites, and even to create shortcuts between these locations [66,67]. By keeping a low number of flowers, we ensured the number of transitions to remember would be low so that this hypothesis was reasonable.

Positive reinforcement was implemented in models 1[+] and 3[+/-]. It occurred when a bee used a transition leading to a rewarding flower. The probability of using this transition was then multiplied by 1.5, then normalized among other transition probabilities to ensure that all sum up to 1 and no single probability goes beyond a value of 1, as in Reynolds *et al.* [23]. This positive reinforcement is based on the well-known tendency of bumblebees to return to nectar-rewarding places through appetitive learning [68]. Negative reinforcement was implemented in models 2[−] and 3[+/-]. It occurred when a bee used a transition leading to a non-rewarding flower. The bee reduced the probability of using that transition by multiplying it by 0.75 (here also rescaling the probabilities after application of the reinforcement). This negative reinforcement rule was based on the tendency of bumblebees to reduce their frequency of revisits to unrewarded flowers with experience [21]. We applied a lower value to negative reinforcement because bees are much more effective at learning positive stimuli (visits to rewarding flowers) than negative stimuli (visits to non-rewarding flowers) (review in [69]). Sensitivity analyses of these two parameters show that increasing positive and/or negative reinforcement increases the speed and level of resource partitioning (S2 Text). However, only positive reinforcement has a significative effect on route similarity (Fig C in S2 Text).

## Competitive interactions

We implemented competitive interactions between foragers in the form of exploitation and interference (Fig 1). Exploitation competition occurred when a bee landed on a flower whose nectar reward had already been collected by another bee. If the flower was empty, the probability to reuse the transition was either left unchanged (model 1[+]) or decreased (negative reinforcement, models 2[−] and 3[+/-]). Interference competition occurred when two bees arrived simultaneously on a flower. Only one bee could stay on the flower and access the potential nectar reward with a random probability (p = 0.5). After the interaction, the winner bee took the reward if there was one. The loser bee reacted as it would for an empty flower. To our knowledge, there is no empirical data suggesting that bees would react differently to these types of competitive interactions. Therefore, we made the parsimonious assumption that the effect was the same.

## Data analyses

All analyses were performed in R version 3.3 [70].

**Simulations with one forager.** For each model, we compared the results of the simulations to the reference observational data, either quantitatively (for [22]) or qualitatively (for [13]; S3 Text). We stopped the simulations after the bees completed a number of foraging bouts matching the maximum observed during the experimental conditions of the published data (37 foraging bouts in [22]; 61 foraging bouts in [13]). We ran 500 simulations for each model and we estimated how models fitted the experimental data using two main measures:

i. the quality of each route, *QL*, calculated as:

$$QL = \frac{\frac{F^2}{d}}{QL_{opt}} \tag{2}$$

Where $F$ is the number of rewarding flowers visited during a foraging bout and $d$ is the net length of all transition movements travelled during the foraging bout. $QL$ is standardized between 0 and 1 by the quality of the optimal route in each array $QL_{opt}$ (shortest possible route to visit all 5 flowers).

ii. a similarity index $SI_{ab}$ between flower visitation sequences experienced during two consecutive foraging bouts $a$ and $b$ as follows:

$$SI_{ab} = \frac{s_{ab}}{2l_{ab}} \qquad (3)$$

Where $s_{ab}$ represents the number of flowers in transitions found in both sequences, and $l_{ab}$ the length of the longest flower visitation sequence between $i$ and $j$ multiplied by 2 to make sure that $SI_{ab} = 1$ occurs only when two consecutive sequences sharing the same transitions also have the same length (more details and examples in S5 Text).

We applied generalized linear mixed effect models (GLMM) with binomial error, using the g*lmer* function in 'lme4' package [71], to assess whether the estimated trends across foraging bouts for $QL$ and $SI_{ab}$ obtained from model simulations with one forager differed from trends obtained from experimental data. In each model, we used a random structure to account for the identity of bees.

**Simulations with two foragers.** We generated 10 arrays of flowers for each of the three types of environments (one patch, two patches and three patches) and ran 100 simulations for each of the three models (9000 simulations in total). We compared the simulation outcomes of the models using four measures:

i. the frequency at which each bee experienced exploitation competition (i.e. flower visits when the reward has already been collected) and interference competition (i.e. flower visits when two bees encounter on the flower).

ii. the similarity index $SI_{ab}$ between successive foraging bouts by the same bee.

iii. the degree of resource partitioning among bees, based on network modularity $Q$ [21,72]. $Q$ is calculated using the *computeModules* function implemented in the R package 'bipartite' [73] using the *DIRTLPAwb+* algorithm developed by Beckett [74]. Although $Q$ ranges between 0 (the two bees visit the same flowers) and 1 (the two bees do not visit any flower in common), the comparison of modularity between networks requires normalisation because the magnitude of modularity depends on network configuration (e.g., total number of flower visits) [74,75]. For each network, we calculated:

$$Q_{norm} = \frac{Q}{Q_{max}} \qquad (4)$$

where $Q_{max}$ is the modularity in a rearranged network that maximizes the number of modules [72].

iv. an index of collective foraging efficiency, $QL_{group}$, computed for each foraging bout $b$, to estimate the collective efficiency of all foraging bees, as:

$$QL_{group,b} = \frac{\sum_{p=1}^{n} QL_{p,b}}{QL_{optimal}} \qquad (5)$$

where $QL_{p,b}$ is the route quality of the individual $p$ during bout $b$, $n$ the number of bees, and $QL_{optimal}$ is the maximum value of all the possible sums of individual route qualities.

$QL_{optimal}$ was calculated in each environment by computing all possible combinations of two routes visiting five flowers each and extracting the combination with the highest quality.

To assess whether the trends across foraging bouts obtained from simulations with two bees differed between models (Fig 1) and types of environments (S2 Fig), we applied GLMMs for each of the following response variables: (*i*) frequency of competition types (Poisson error distribution), (*ii*) $SI_{ab}$ (Binomial error distribution), (*iii*) $Q_{norm}$ (Binomial error distribution) and (*iv*) $QL_{group}$ (Binomial error distribution). In each model, we used a random structure to account for bee identity nested in flower arrays (i.e. 100 simulations of each spatial array for each model). To statistically compare the trends across foraging bouts, we estimated the marginal trends of the models, as well as the 95% confidence intervals of the mean using the *emtrends* function in 'emmeans' package [76]. When the 95% confidence intervals of the estimated trends included zero, the null hypothesis was not rejected. Statistical models were run using the g*lmer* function in 'lme4' package [71].

## Supporting information

**S1 Fig. Experimental Flower Arrays.** Arrays of artificial flowers (grey circles) and the colony nest (black pentagons) used to obtained the experimental datasets. A. Regular pentagon, modified from Lihoreau *et al.* [22]. B. Narrow pentagon, modified from Woodgate *et al.* [13].
(TIF)

**S2 Fig. Simulated Flower Arrays.** Examples of simulated environments. Spatial distribution of 10 flowers (grey circles) and a colony nest (black pentagon) in three types of environments defined by different levels of flower patchiness. A flower patch was characterized by: 1) a uniform distribution of flowers, 2) a lower distance between flowers within the patch than between all flowers from different patches (see details in methods).
(TIF)

**S1 Text. ODD Protocol.**
(DOCX)

**S2 Text. Sensitivity analysis of positive and negative reinforcements.**
(DOCX)

**S3 Text. Qualitative comparison between simulations and observations in the narrow pentagon.**
(DOCX)

**S4 Text. Predictions with more than two bees.**
(DOCX)

**S5 Text. Supplementary information on the similarity index.**
(DOCX)

**S6 Text. Details on the movement probability matrix.**
(DOCX)

**S1 Video. Animation of a model simulation with 5 bees.** Example of simulation of five bees foraging in an environment with one patch of 50 flowers. Both positive and negative reinforcement rules are implemented (model 3[+\-]). Bees performed 100 foraging bouts.
(MP4)

## Acknowledgments

We thank Joe Woodgate for sharing raw flower visitation data of Woodgate *et al.* (2017) [13]. We also thank Jerome Buhl and Andy Reynolds for fruitful discussions on modelling. We are grateful to Matthias Becher and Mickaël Henry for useful comments on a previous version of this manuscript.

## Author Contributions

**Conceptualization:** Thibault Dubois, Cristian Pasquaretta, Mathieu Lihoreau.

**Formal analysis:** Thibault Dubois, Cristian Pasquaretta.

**Software:** Thibault Dubois, Jacques Gautrais.

**Supervision:** Mathieu Lihoreau.

**Writing – original draft:** Thibault Dubois, Cristian Pasquaretta, Mathieu Lihoreau.

**Writing – review & editing:** Thibault Dubois, Cristian Pasquaretta, Andrew B. Barron, Jacques Gautrais, Mathieu Lihoreau.

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
