## [Decision Letter · Decision Letter 0]

29 Mar 2021

Dear Mr. DUBOIS,

Thank you very much for submitting your manuscript "A model of resource partitioning between foraging bees based on positive and negative associations" for consideration at PLOS Computational Biology.

As with all papers reviewed by the journal, your manuscript was reviewed by members of the editorial board and by several independent reviewers. In light of the reviews (below this email), we would like to invite the resubmission of a significantly-revised version that takes into account the reviewers' comments.

We cannot make any decision about publication until we have seen the revised manuscript and your response to the reviewers' comments. Your revised manuscript is also likely to be sent to reviewers for further evaluation.

Sincerely,

Phillip Staniczenko

Guest Editor

PLOS Computational Biology

Stefano Allesina

Deputy Editor

PLOS Computational Biology

Reviewer's Responses to Questions

**Comments to the Authors:**

Reviewer #1: This study explores how two possible mechanisms – positive reinforcement and negative reinforcement – might cause the development of traplines and resource partitioning in foraging bumblebees. The authors have developed an individual based model in which the probability of movement between any two flowers is given in a matrix for each bee. The bee moves from flower to flower and each time a bee visits a flower, the probability associated with that movement is updated according to the mechanism(s) used. The authors compare three versions of their model: with positive reinforcement only, with negative reinforcement only, and with both types of reinforcement.

The authors find that, for two bees foraging at the same time, these mechanisms have different effects in different environments (number of patches). In two or three patch environments, positive reinforcement alone allows for traplining but not resource partitioning, while negative reinforcement alone allows for resource partitioning but not traplining. When both are combined, traplining and resource partitioning result. In a single patch environment, negative reinforcement alone is not sufficient for either traplining or resource partitioning. Positive reinforcement alone or positive reinforcement and negative reinforcement combined allow for both traplining and resource partitioning.

I find the study to be well grounded in the literature on trapline formation, learning, and navigation in bumblebees. The authors make use of two previously published experimental studies (of trapline formation in bumblebees foraging on artificial arrays of flowers) for model validation. I have however, provided (below in the detailed comments) a few additional bee movement modelling references that the authors may use to support their claims and better situate their work within the modelling literature.

General Comments

I find this to be an interesting and compelling model that adheres to the modelling principles of parsimony specificity. The authors avoid making the model needlessly complex and include only behavioural details that are relevant to the research question and well enough understood to be confidently modelled, but they include enough detail of the relevant behaviours to be able to fully address the research question. To my knowledge, there are no other bee movement models which address resource partitioning, and indeed the number of bee models of any learning behaviour at all is limited, making this an excellent addition to the literature.

I find, however, that the introduction of the paper and the presentation of the results sets up the reader to expect a greater treatment of traplining (route similarity) in the discussion. The combination of the effects of the positive and negative reinforcements in different landscapes with both resource partitioning and route similarity (as I have summarized above) presents a more complex, and thus more interesting, picture of the results of the model. I would suggest that the authors include more to this effect in the discussion section.

Detailed Comments

1. Line 94-95. What is vector-based learning? If I understand correctly it refers to the fact that the bees are remembering the vectors between significant locations, rather than the entire path they have followed in each foraging bout. This should be clarified here.

2. Throughout lines 84-113, I found myself getting confused about route-based learning and vector-based learning. In Lines 101 and 110 you use the term ‘route learning’ in reference to your model, which uses vector-based learning. Perhaps ‘trapline formation’ would be a better term here?

3. Line 96-98. I suggest a few bee population/movement modelling papers which could provide broader support for your claims here than just relying on the BEEHAVE suite of models, although you may be able to find others.

H. Qu and F. Drummond. Simulation-based modelling of wild blueberry pollination. Computers and Electronics in Agriculture, 144:94–101, 201

J. Everaars, J. Settele, and C. F. Dormann. Fragmentation of nest and foraging habitat affects time budgets of solitary bees, their fitness and pollination services, depending on traits: Results from an individual-based model. PLOS One, 13(2):e0188269, 2018.

S. A. Rands. Landscape fragmentation and pollinator movement within agricultural environments: a modelling framework for exploring foraging and movement ecology. PeerJ, 2:e269, 2014.

4. Line 101. Do you mean comparing the models to each other? Or to data? Both?

5. Lines 137-139. You explain later in the manuscript that the sample size in the narrow pentagon array study was too small to allow for quantitative comparison, but I feel that that information should be presented here, otherwise the reader is left wondering why only qualitative comparison was possible.

6. Lines 160-169. Here I wondered, if you decrease the positive reinforcement does that make the model match the data better? Since this is the main validation of the model, I would like to see a greater discussion of the discrepancy between the model and the data. This would easily be done by incorporating the discussion from S3 into the main manuscript.

7. Lines 188-192 this decrease/increase in non-rewarding flower visits is accompanied by a corresponding decrease/increase in total number of flower visits, correct? (and lines 208-211 for encounters on flowers)

8. Throughout the results, I kept missing the ‘and model 3’ in between all of the parenthetical GLMM results and getting confused that I had missed something. I don’t have a suggestion for a way to avoid this, but if you can think of one, I’m sure future readers would appreciate it!

9. Lines 395-400. Are flowers in the same patch really 20m apart? This seems quite far!

10. How realistic is it to use exactly the number of flowers for the bees to fill their crops? Does this have an affect on their behaviour?

11. The model implies that the bees are able to remember the locations and distances of all of the flowers in their environment. How realistic is this?

12. Line 428. This means probabilities greater than 1 can occur. Are any values greater than 1 that result from the multiplication by 1.5 set to 1?

13. Line 437-438. Did you look at the effect on route similarity? Or any of the other metrics shown in figure 3? Although the focus of the paper is on resource partitioning, I would be interested to hear about the effects on route similarity as well as resource partitioning (and maybe the rest could be included in the supporting information).

14. Figure 1 appears to be missing an arrow and some yes/no statements for the ‘is the maximum flight distance reached?’ box.

Reviewer #2: An individual-based model is presented that represents foraging of bumblebees and addresses the question how bumblebees develop „traplines“, i.e. preferred sequences of flowers they tend to visit during a foraging bout. While models on trapline emergence exist for individual bees, the question how two (or more) bees develop non-overlapping traplines and thereby avoid competition and increase resource partitioning has not been addressed with simulations yet.

The settings of the model are based on experiments with individual foragers, i.e. artificial landscapes with five (or 15) flowers, which refill their nectar store after each foraging bout (i.e., all bees of the current bout returned to their nest).

The question addressed is interesting. It is part of the overall topic of optimal foraging and the role of learning and cognitive capacities involved. The manuscript is well written and well-organized. Overall I think this is an interesting study that provides some important new insights, in particular that both, negative and positive reinforcement learning, are required to explain the emergences of non-overlapping traplines.

I am not an expert in bees or foraging theory, but in individual-based modelling, my comments therefore reflect a modeller’s perspective.

I like the model, but of course there are always assumptions that can be debated. Here, a key assumption is that, effectively, the bees are „omniscient“, they know the position and distance to all flowers in the landscape via their distance matrix. Of course, this is not the case in reality but rather a technical proxy. In reality, bees have a perception range, which means they certainly cannot know the flowers in clumps of flowers hundreds of meters away.

I understand from reading the ODD description, that within flower patches for the positioning of flowers „perception range“ played a role, but I think the authors should explain the underlying rationale in more detail, in particular when it comes to the landscape with three flower patches. I guess the main argument for assuming the distance matrix is a good proxy is that by taking the distance to the current position into account the probability of finding that flower by chance is taken into account, but this is something that could be tested. Why, for example, did you use the square of the distances? Did you try different exponents?

Thus, the model description should explain and justify the distance matrix in more detail, for example by: presenting a landscape and the typical distance matrix of a bee, before and after learning; plotting the matrix elements over distance to get a feeling of how much less likely it is to find other patches, etc.

I am not saying the distance matrix is a flawed design, it is actually quite elegant, but, as it goes with proxies, you easily get used to using it and forget asking about its implicit assumptions, which might, for certain configurations, be quite unrealistic.

Further comments:

- Line 142: „GLMM …“ – I find these lengthy references to statistical metrics not very interesting by themselves, but then there are more than 50 of them in the main text. Yes, it makes everything look more „scientific“, but I think it would be sufficient to include this information in a Table in the Supplement, it would make the main text easier to read.

- I read the main text before looking at the figures and was slightly challenged by distinguishing model 1, 2 and 3. Perhaps say „model 1+, model 2-, and model 3+-“ to indicate also within the text that they about positive, negative, and combined re-enforcement? Just an idea.

- Line 191: „bees tended to increase their visits to non-rewarding flowers“ – since it is difficult to develop intuitions about how the distance matrix and its change due to learning, actually work, one cannot develop any institution about the reason for this observation. Still, the authors should be able to give a hint WHY this is so.

- Line 211: „increased their frequency“ – same as in previous comment.

- I like the outline of possible routes for making the model more „realistic“, but one important element is missing: most plants have more than one flower, how would this change the story? (Not that much, I guess, but being not a bee expert, I don’t have a good intuition here).

- Line 358: „pushed away“ – in terms of modulating the distance matrix, there is no difference between exploitation and interference competition, but could there be a difference in reality, with direct encounters with other bees having a stronger impact?

- Speaking of „movement vectors“ is slightly confusing, as you are actually not using vectors in the model the way they are use in other foraging models where animals remember the position of rewarding sites. I also would not speak of „vector navigation“ as this indicates that, like in the models I mentioned (e.g. van Moorter et al. 2009, Milles et al. 2020), in models using vector navigation vectors are actually added to determine the direction for the next movement step. In your model, there is not such navigation or movement.

Supplement

Overall, the ODD model description is fine, but here still a few comments:

- „Purpose and patterns“ – you did not list the patterns used to claim that model output is realistic enough for the purpose of the model.

- Line 29 „Internal parameters“ – see my comments on the rationale underlying the distance matrix.

- Line 147 „Initialization“ – this is about initializing the model entities and their state variables, not about setting model parameter, which has been intermingled here as well.

- Line 177 „Input data“ – this is common misunderstanding of the ODD element which has been reported on both in Grimm et al. 2010 and 2020: this section is about the input of environmental variables, for example via data files, not about user input! This section should be moved to the begin of the Submodels section. You don’t have any “Input DATA”.

- Line 306 – I understand that indFlowerOutcome is an auxiliary state variable of the bees with all elements set to 1 after each foraging bout?

References

- Milles, A., Dammhahn, M., & Grimm, V. (2020). Intraspecific trait variation in personality‐related movement behaviour promotes coexistence. Oikos, 129(10), 1441-1454.

- Van Moorter, B., Visscher, D., Benhamou, S., Börger, L., Boyce, M. S., & Gaillard, J. M. (2009). Memory keeps you at home: a mechanistic model for home range emergence. Oikos, 118(5), 641-652.

**Have all data underlying the figures and results presented in the manuscript been provided?**

Reviewer #1: Yes

Reviewer #2: Yes

PLOS authors have the option to publish the peer review history of their article (what does this mean?). If published, this will include your full peer review and any attached files.

Reviewer #1: No

Reviewer #2: No
---

## [Decision Letter · Decision Letter 1]

7 Jul 2021

Dear Mr. DUBOIS,

We are pleased to inform you that your manuscript 'A model of resource partitioning between foraging bees based on learning' has been provisionally accepted for publication in PLOS Computational Biology.

Best regards,

Phillip Staniczenko

Guest Editor

PLOS Computational Biology

Stefano Allesina

Deputy Editor

PLOS Computational Biology

The two original reviewers have now seen your revised manuscript and both are supportive of publication. I agree with their assessments. Also, please consider implementing the two latest suggestions from Reviewer 1.

Reviewer's Responses to Questions

**Comments to the Authors:**

Reviewer #1: Thank you for your excellent responses to my comments, questions, and suggestions. I just have two remaining comments.

1) I think your statement in lines 99-100 that “current models of bee populations do not consider individual specificities of movements based on learning and memory” is a little bit too strong. Bumble-BEEHAVE, for example, does include memory in that bees in this model are more likely to return to a previously visited rewarding patch, although movement is not a focus of the model. I would suggest softening this statement by saying “current models of bee populations often do not consider…” or something similar.

2) In your edited lines 303-305, you say “This assumption is comforted by the fact that both reinforcements leads to a greater resource partitioning and a higher collective foraging efficiency.” I don’t think “comforted” is the word you meant to use. Maybe “confirmed,” but I’m not quite sure, so please check this.

Reviewer #2: The authors thoroughly addressed all of my comments and suggestions, in particular my question about the distance matrix. With this, this interesting work is ready to be published.

**Have the authors made all data and (if applicable) computational code underlying the findings in their manuscript fully available?**

Reviewer #1: None

Reviewer #2: Yes

PLOS authors have the option to publish the peer review history of their article (what does this mean?). If published, this will include your full peer review and any attached files.

Reviewer #1: No

Reviewer #2: No

---

## [Editor Report · Acceptance letter]

23 Jul 2021

PCOMPBIOL-D-21-00192R1 

A model of resource partitioning between foraging bees based on learning

Dear Dr DUBOIS,

I am pleased to inform you that your manuscript has been formally accepted for publication in PLOS Computational Biology. Your manuscript is now with our production department and you will be notified of the publication date in due course.

With kind regards,

Olena Szabo
